# The percentage of CD39+ monocytes is higher in pregnant COVID-19+ patients than in nonpregnant COVID-19+ patients

**A. Cérbulo-Vázquez**[1]\*, **M. García-Espinosa**[2], **J. C. Briones-Garduño**[3], **L. Arriaga-Pizano**[4], **E. Ferat-Osorio**[5], **B. Zavala-Barrios**[3], **G. L. Cabrera-Rivera**[4], **P. Miranda-Cruz**[4], **M. T. García de la Rosa**[4], **J. L. Prieto-Chávez**[4,6], **V. Rivero-Arredondo**[4], **R. L. Madera-Sandoval**[4], **A. Cruz-Cruz**[4], **E. Salazar-Rios**[4], **M. E. Salazar-Rios**[4], **D. Serrano-Molina**[4], **R. C. De Lira-Barraza**[4], **A. H. Villanueva-Compean**[7], **A. Esquivel-Pineda**[7], **R. Ramirez-Montes de Oca**[7], **F. Caldiño-Soto**[8], **L. A. Ramírez-García**[9], **G. Flores-Padilla**[7], **O. Moreno-Álvarez**[9], **G. M. L. Guerrero-Avendaño**[10], **C. López-Macías**[4,11]\*

1 Departamento de Medicina Genómica, Hospital General de México "Dr. Eduardo Liceaga", Ciudad de México, México, 2 Servicio de Complicaciones de la Segunda Mitad del Embarazo, UMAE Hospital de Gineco-Obstetricia No. 4 "Dr. Luis Castelazo Ayala". Instituto Mexicano del Seguro Social (IMSS), Ciudad de México, México, 3 Dirección de Medicina Aguda, Diagnóstico y Tratamiento, Hospital General de México "Dr. Eduardo Liceaga", Ciudad de México, México, 4 Unidad de Investigación Médica en Inmunoquímica, UMAE Hospital de Especialidades, Centro Médico Nacional Siglo XXI. IMSS, Ciudad de México, México, 5 División de Investigación, UMAE Hospital de Especialidades, Centro Médico Nacional Siglo XXI. IMSS, Ciudad de México, México, 6 Centro de Instrumentos, UMAE Hospital de Especialidades, Centro Médico Nacional Siglo XXI. IMSS, Ciudad de México, México, 7 Medicina Interna, UMAE Hospital de Especialidades, Centro Médico Nacional Siglo XXI. IMSS, Ciudad de México, México, 8 División Obstetricia, UMAE Hospital de Gineco-Obstetricia No. 4 "Dr. Luis Castelazo Ayala", Instituto Mexicano del Seguro Social (IMSS), Ciudad de México, México, 9 Dirección Médica, UMAE Hospital de Gineco-Obstetricia No. 4 "Dr. Luis Castelazo Ayala", Instituto Mexicano del Seguro Social (IMSS), Ciudad de México, México, 10 Dirección General, Hospital General de México "Dr. Eduardo Liceaga", Ciudad de México, México, 11 Visiting Professor of Immunology, Nuffield Department of Medicine, University of Oxford, Oxford, United Kingdom

\* cerbulo@unam.mx (ACV); constantino@sminmunologia.mx, constantino.lopez@imss.gob.mx (CLM)

**Data Availability Statement:** All relevant data are within the manuscript and its Supporting information files.

## Abstract

Current medical guidelines consider pregnant women with COVID-19 to be a high-risk group. Since physiological gestation downregulates the immunological response to maintain "maternal-fetal tolerance", SARS-CoV-2 infection may constitute a potentially threatening condition to both the mother and the fetus. To establish the immune profile in pregnant COVID-19+ patients, a cross-sectional study was conducted. Pregnant women with COVID-19 (P-COVID-19+; n = 15) were analyzed and compared with nonpregnant women with COVID-19 (NP-COVID-19+; n = 15) or those with physiological pregnancy (P-COVID-19-; n = 13). Serological cytokine and chemokine concentrations, leucocyte immunopheno-types, and mononuclear leucocyte responses to polyclonal stimuli were analyzed in all groups. Higher concentrations of serological TNF-α, IL-6, MIP1b and IL-4 were observed within the P-COVID-19+ group, while cytokines and chemokines secreted by peripheral leu-cocytes in response to LPS, IL-6 or PMA-ionomicin were similar among the groups. Immu-nophenotype analysis showed a lower percentage of HLA-DR+ monocytes in P-COVID-19+ than in P-COVID-19- and a higher percentage of CD39+ monocytes in P-COVID-19+ than in NP-COVID-19+. After whole blood polyclonal stimulation, similar percentages of T cells

**Funding:** CLM awarded a Project No. 313494 by the Mexican National Research Council (CONACyT). https://conacyt.mx/ The funders had no role in study design, data collection and analysis, decision to publish, or preparation of the manuscript.

**Competing interests:** All authors declare no competing interests.

and TNF+ monocytes between groups were observed. Our results suggest that P-COVID-19+ elicits a strong inflammatory response similar to NP-COVID19+ but also displays an anti-inflammatory response that controls the ATP/adenosine balance and prevents hyperinflammatory damage in COVID-19.

## Introduction

Severe acute respiratory syndrome coronavirus (SARS-CoV) and Middle East respiratory syndrome coronavirus infections result in high mortality among pregnant women (25% and 27%, respectively) [1]. In 2019, a new coronavirus called SARS-CoV-2 appeared and became a new high-risk virus because SARS-CoV-2 viral infections lead to a powerful cell and humoral immune response in pregnant women and increase fetal morbidity and mortality [2–4]. Additionally, pregnant women exhibit higher mortality rates associated with viral infections than the general population [5, 6]. The immune response in pregnant women is mediated by many cellular and humoral mechanisms [7, 8], resulting in a unique biological scenario to face a new virus, as in COVID-19.

Comorbidities associated with critical COVID-19 are highly prevalent in Mexico in both the general population and pregnant women [3, 4, 9, 10]; however, the clinical presentation of COVID-19 seems to be similar in both the general population and pregnant women [3, 11]. Current data are not sufficient to determine whether vertical transmission is possible, how large the potential infection is by asymptomatic pregnant women to the general population, or whether COVID-19 leads to an increase in postpartum mortality. In addition, the epidemiological and clinical characteristics and immune profile of pregnant women with COVID-19 have been poorly explored. As it is in the general population, lymphopenia is also reported in pregnant women with COVID-19 [12]. Parameters such as neutrophil count are useful for COVID-19 diagnosis [13]. Furthermore, an increased neutrophil/lymphocyte ratio has been associated with fatal outcomes in COVID-19 patients [14–17]. Several studies indicate that leucocyte count is necessary for the initial evaluation of COVID-19 patients; however, a detailed phenotypic analysis of leucocytes could improve our knowledge of SARS-CoV-2 infection, especially in pregnant women.

Several humoral components of the immune response are involved in COVID-19; among them, inflammatory cytokines/chemokines have been observed in high serum concentrations in the general population, and the higher the concentration is, the worse the clinical status [18, 19]. Cytokines and chemokines are key regulators in the physiological immune response and in COVID-19. SARS-CoV induces low expression of IFN-α, IFN-β and IL-10, moderate expression of TNF-α and IL-6, and high expression of CCL3, CCL5, CCL-2 and CXCL10 [20, 21]. In addition, SARS-CoV protein S induces CCL2 and CXCL8 synthesis *in vitro* [22, 23]. COVID-19 patients show higher concentrations of IL-2, IL-7, IL-10, G-CSF, IP10 (CXCL10), MCP-1 (CCL2), MIP1a (CCL3) and TNF-α than non-COVID-19 patients [18]. The sources of these cytokines/chemokines are diverse and could include lung epithelial cells, endothelial cells, and leucocytes [24–26].

The COVID-19-derived inflammatory response in pregnancy could disturb the delicate immune balance in cellular components. Some surface molecules on cells can modulate leucocyte activation and function and, in pregnancy, also trigger regulator systems that may prevent both fetal rejection and the inflammatory response. CD39 and CD73 are ectonucleotidases that metabolize ATP to adenosine, which drives immune profiles from proinflammatory to anti-inflammatory [27]. Additionally, CD39 downregulation is associated with poor

pregnancy outcomes [28]. Analysis of CD39 and CD73 in COVID-19 patients showed a decrease in CD8+CD73+ T cells, and NKT cells correlated with ferritin levels and were potential useful prognostic markers in COVID-19 [29]. Additionally, lower plasma ATP and adenosine levels were identified in mild and severe COVID-19 patients, higher frequencies of CD4+CD39+ cells in severe COVID-19 patients, and diminished frequencies of CD4+CD73+ and CD8+CD73+ cells in severe COVID-19 patients compared to mild COVID-19 patients and controls [30]. However, no evidence of the CD39/73 system has been reported in pregnant women with COVID-19.

To analyze the immune profile in P-COVID-19+, a cross-sectional study was conducted. Our analysis includes a) the immunophenotype of lymphocytes and monocytes expressing activation markers, b) serum cytokine/chemokine concentration, and c) the proportion of cytokine-producing leucocytes in response to polyclonal stimuli. Herein, we explored cellular and humoral characteristics that may improve the understanding of the immunopathophysiology of pregnant women with COVID-19.

## Material and methods

### Patients

This study was conducted by the "Servicio de Ginecología y Obstetricia" at the Hospital General de Mexico "Dr. Eduardo Liceaga" in conjunction with the Unidad de Investigación Médica en Inmunoquímica (UIMIQ) at the Hospital de Especialidades, Centro Médico Nacional Siglo XXI, and the Servicio de complicaciones de la segunda mitad del embarazo, División Obstetricia, UMAE Hospital de Gineco-Obstetricia No. 4 "Dr. Luis Castelazo Ayala". The study was evaluated by the National Committee for Scientific Research (CNIC) with the following approval numbers: Research project: DI / 20112/04/45, and R-2020-785-095. After obtaining a signed informed consent letter, forty-four women were enrolled and assigned to one of three groups: a) NP-COVID-19+, n = 15, b) P-COVID-19+, n = 15, and c) P-COVID-19-, n = 13. The COVID-19 diagnosis was based on clinical characteristics [31], and SARS-CoV-2 viral infection was confirmed by specific reverse transcription–polymerase chain reaction. Comorbidities and clinical features were recorded.

### Blood sample collection

Our study is in accordance with the World Medical Association's Declaration of Helsinki. After a patient signed the informed consent letter, blood specimens were collected in silicone-coated tubes (EDTA or heparinized tubes, BD Vacutainer, N. J, USA). Samples were processed immediately after collection.

### Leucocyte immunophenotyping

Whole blood samples (50 μL) were incubated with titrated volumes of antibodies according to the following panel: anti-CD45-PerCP (Clone:HI30), anti-CD3-AF647 (Clone:UCHT1), anti-CD14-PECy7 (Clone:M5E2), anti-CD16-FITC (Clone:3G8), anti-CD19-APC/Cy7 (Clone: HIB19), anti-CD73-PE (Clone:AD2), anti-CD39-BV421 (Clone:A1), anti-CD4-APC/Cy7 (Clone:OKT4), anti-CD8-PE/Dazzle594 or BV510 (Clone:SK1), anti-CD69-BV421 (Cone: FN50), anti-HLA-DR-AF488/PE/Dazzle594 (Clone:L243), and dead cells with Zombie Aqua fixable viability kit (all reagents from BioLegend, San Diego, CA). After 15 minutes of incubation, erythrolysis was performed using FACS™ Lysing Solution (Cat. 349202, BD, San Jose, CA, USA). Samples were washed once with 1x PBS (1,500 rpm, 5 minutes, 4°C) and resuspended in PBS (100 μL). At least 30,000 leucocytes (CD45+ cells) were acquired in a FACSAria IIu

flow cytometer (BD Biosciences, San José, CA, USA). The FACS files were analyzed with Infinicyt™ software 1.8 (Cytognos, Salamanca, Spain). Single cells were defined with an FSC-A vs. FSC-H plot, and leucocytes were identified using an SSC vs. CD45 plot. Lymphocytes were gated as $SSC^{low}FSC^{low}CD45^{++}CD14^{-}$, monocytes as $SSC^{mid}FSC^{mid}CD45^{+}CD14^{+}$, and neutrophils as $SSC^{mid}FSC^{mid}CD45^{+}CD16^{+}$. Lymphocyte subtypes were identified according to CD19 + (B cells) and CD3+ (T cells), which could be CD4+ (Th) or CD8+ (Tc). The percentages and mean fluorescence intensities (MFIs) of HLA-DR, CD69, CD39, CD73, CD32 and CCR5 were calculated.

## Polyclonal stimulation

Whole heparinized blood ($10x10^{9}$ cells/mL/well) was incubated (4 hours at 37°C with 5% $CO_2$) alone in 24-well culture plates (Cat 13485, Costar, NY, USA), with human recombinant IL-6 (human rIL-6, 100 ng/mL), or with *Escherichia coli* O55:B5 lipopolysaccharide (LPS 250 ng/mL, Cat. L2880, Sigma Aldrich, St. Louis, MO, USA) or with phorbol myristate acetate/ ionomycin (PMA 50 ng/mL, ion 1 mg/mL). The supernatant was recovered and stored at -20°C until cytokine and chemokine assessment. Independently, some of the samples were incubated with the stimuli described above in the presence of brefeldin-A (Cat. 420601, BioLegend, San Diego, CA, USA). Afterward, intracellular phenotyping was performed.

## Intracellular cytokine detection

After cell culture, whole blood samples ($10x10^{5}$ cells/mL) were incubated with the following panel: antibodies from BioLegend, San Diego, CA, USA: anti-CD45-PerCP (Clone: HI30), anti-CD3-AF647 (Clone: UCHT1), anti-CD14-PECy7 (Clone: M5E2), anti-CD4-APC/Cy7 (Clone: OKT4), anti-CD8-PE/Dazzle594 or BV510 (Clone: SK1). After 15 min in the dark, blood was washed once with PBS (1 mL) by centrifugation at 900×g for 5 min at room temperature (RT); then, Fixation buffer was added (100 μL, Cat: 420801, BioLegend, San Diego, CA, USA), and the samples were incubated for 20 min. Then, samples were washed twice with 1 mL of Intracellular Staining Perm Wash buffer (Cat: 421002, BioLegend, San Diego, CA, USA); after the second wash, they were mixed with monoclonal antibodies against cytokines from BioLegend, San Diego, CA, USA: anti-TNFα-BV421 (Clone: MAb11), anti-IL-6-PE (MQ2-13A5), anti-IL-1β-FITC (Clone:JK1B-1), anti-IFNγ-BV421 (Clone:4S. B3), anti-IL-8a-PE (Clone: E8N1). For the exclusion of dead cells, a Zombie Aqua fixable viability kit (BioLegend, San Diego, CA, USA) was added and incubated for 30 min at RT. Last, the mixture was washed once with PBS. At least 30,000 events were acquired in a FACSAria IIu (BD, San Jose CA) flow cytometer. Analysis was performed using Infinicyt™ Software 1.8.

## Soluble cytokine/chemokines quantification

Serum or cell culture supernatant was analyzed as follows: cytokines (IL-2, IL-4, IL-6, IL-10, TNF-α, IFN-γ, and IL-17a) and chemokines (CXCL8/IL-8, CXCL10/IP-10, CCL11/Eotaxin, CCL17/TARC, CCL2/MCP-1, CCL5/RANTES, CCL3/MIP-1a, CXCL9/MIG, CXCL5/ENA-78, CCL20/MIP-3a, CXCL1/GROa, CXCL11/I-TAC and CCL4/MIP-1b) were determined using bead-based immunoassays (CBA kit, Cat. 560484, BD Pharmingen, San Diego, CA, USA; and LEGENDplex, Cat. 740003, BioLegend, San Diego, CA, USA, respectively). Log-transformed data were used to construct standard curves fitted to 10 discrete points using a 4-parameter logistic model. The concentration of each cytokine/chemokine was calculated using interpolations of their corresponding standard curves.

## Statistical analysis

Statistical analysis was performed using GraphPad Prism® version 7 software (GraphPad Software, San Diego, CA, USA). Nonparametric ANOVA (Kruskal–Wallis test) with Dunn's post-test was applied. Categorical variables were expressed as percentages (% and compared by Fisher's exact test. A $p<0.05$ was considered statistically significant.

## Results

We compared P-COVID-19+, NP-COVID-19+ and clinically healthy pregnant women (P-COVID-19-) to assess the immune profile. Table 1 shows the demographic and clinical features. Pregnant women with or without COVID-19 had similar maternal and gestational ages. Additionally, the frequency of comorbidities (diabetes mellitus and systemic arterial hypertension) was similar among the groups. Some clinical characteristics, such as heart rate and serological lactate dehydrogenase (LDH) concentration, were higher in the P-COVID-19+ patients than in the P-COVID-19- patients (p = 0.048 and p = 0.005, respectively). No significant

**Table 1. Clinical and laboratory characteristics.**

| | NP-COVID-19+ (n = 9–15)a | P-COVID-19+ (n = 8–15)b | P-COVID-19- (n = 13)c | *p* |
|---|---|---|---|---|
| Age (years) | 34.1±7 | 28±7.1 | 25.6±5.7 | a vs. c 0.006 |
| BMI | 31.6±6.9 | 30.0±6.3 | 28.0±3.5 | **0.356** |
| Respiratory Rate (breaths per min) | 23±3.8 | 22.1±6.8 | 18.6±1.1 | a vs. c 0.005 |
| Heart Rate (beats per min) | 91.2±25.1 | 97.3±19.6 | 77.7±7.5 | b vs. c 0.048 |
| Temperature (˚C) | 36.9±1.2 | 36.7±0.7 | 36.3±0.2 | **0.081** |
| Mean Arterial Pressure (mmHg) | 159.5±17.4 | 153.2±15.4 | 157.1±13.3 | **0.631** |
| Hemoglobin | 13±2.7 | 12.6±2.0 | 11.9±1.1 | **0.070** |
| Leucocytes (10xmm$^3$) | 8.6±4 | 11.1±3.2 | 8.4±2.3 | **0.140** |
| NLR | 11.8±11.7 | 5.1±1.9 | 3.7±1.2 | a vs. c 0.009 |
| PLT (10xmm$^3$) | 272.3±99.2 | 327.2±218.1 | 232.8±45.4 | **0.548** |
| Glucose (mg/dL) | 122.5±51.8 | 91.1±20.1 | 89.3±12.5 | **0.170** |
| Urea (mg/dL) | 30.9±20.4 | 17.4±5.7 | 18.3±5.9 | a vs. b 0.040 |
| | | | | a vs. c 0.027 |
| Creatinine (mg/dL) | 0.7±0.3 | 1.0±1.1 | 0.5±0.2 | **a vs. c 0.009** |
| LDH (U/L) | 804.9±1389 | 582.8±510.8 | 269.7±90.4 | b vs. c 0.005 |
| PT (seconds) | 15.2±3.3 | 11.5±1.1 | 11±0.4 | a vs. b 0.011 |
| | | | | a vs. c 0.0002 |
| PTT (seconds) | 31.2±4.6 | 30.9±4 | 29.5±3.3 | **0.595** |
| D-Dimer (ng/mL) | 3.9±6.9 | 1513±1474 | 3215±1247 | a vs. b 0.028 |
| | | | | a vs. c <0.0001 |
| Fibrinogen (mg/dL) | 662.7±167.5 | 693.9±186.3 | 566.2±103.4 | **0.195** |
| Oxygen saturation (%) | 88.4±13.3 | 91.1±5.6 | 98.5±0.5 | a vs. c 0.049 |
| Diabetes | 2 | 3 | 1 | **0.614** |
| Hypertension | 1 | 2 | 0 | **0.375** |
| Obesity | 5 | 6 | 0 | 0.037 |
| Gestational age (Weeks) | — | 31.4±5.7 | 35.3±4.8 | **0.069** |
| 2nd trimester | — | 2 | 1 | **Ns** |
| 3rd trimester | — | 13 | 12 | **Ns** |

Non-Pregnant COVID-19+ (NP-COVID-19+), Pregnant COVID-19+ (P-COVID-19+), Pregnant COVID-19- (P-COVID-19-). BMI; Body Mass Index. NLR; Neutrophil/Lymphocyte Ratio. Kruskal–Wallis test and Dunn's multiple comparisons test. Significant $p<0.05$. Mean±SD.

difference was observed between the P-COVID-19+ and P-COVID-19- patients for age, BMI, respiratory rate, body temperature, mean arterial pressure (MAP), hemoglobin, total leucocyte count, neutrophil/lymphocyte ratio (NLR), total platelet count, serum glucose, serum creatinine, partial thromboplastin time or fibrinogen. Diminished oxygen saturation levels were observed in the COVID-19+ pregnant and nonpregnant patients, but without significant differences among them. Regarding D-dimer concentrations, higher levels were observed in the P-COVID-19+ than in the NP-COVID-19+ patients; however, the highest D-dimer concentration was found in those patients with physiological pregnancy. We did not observe differences among groups in total leucocyte count or in subtypes of leucocytes identified by flow cytometry (S1 Table). However, the lowest percentage of lymphocytes was observed in the NP-COVID-19+ patients.

Table 2 shows the frequency of symptoms in those patients with and without COVID-19. The pregnant women with COVID-19 had a similar frequency of symptoms such as cough, myalgia, arthralgia and dyspnea as the NP-COVID19+ group. The most frequent symptoms in the pregnant patients with COVID-19 were cough, myalgia, and arthralgia.

Using flow cytometry, we analyzed the cellular phenotype in the COVID-19 patients. The proportion of leucocytes with an activated phenotype (HLA-DR+ or CD69+) or inflammatory modulators (CD39+ or CD73+) was determined in peripheral blood samples. Fig 1A shows a lower proportion of monocytes HLA-DR+ in the pregnant COVID-19+ patients compared to the pregnant COVID-19- patients (p = 0.013). No differences were observed between the pregnant and nonpregnant COVID-19+ patients (p = 0.481). In contrast, CD4 or CD8 T lymphocyte HLA-DR+ or CD8+CD69+ cells were similar among the groups (S2 Table). Regarding CD39 expression in leucocytes (Fig 1B–1D), the percentage of monocytes CD39+ was significantly higher in the P-COVID-19+ than in NP-COVID-19+ patients (p = 0.007), and higher frequencies of B cells CD39+ (Fig 1C, p = 0.089), and lower T cells CD39+ (Fig 1D, p = 0.111) in the P-COVID-19+ than in NP-COVID-19+ patients.

Regarding CD73 (Fig 1E–1G), we found a lower percentage of CD73+ monocytes in the P-COVID-19+ than in the NP-COVID-19+ patients, without statistical significance (Fig 1E, p = 0.091). Despite pregnancy or COVID-19, similar percentages of CD73+ B and T cells were observed (Fig 1F, p>0.9 and Fig 1G, p = 0.501, respectively).

To explore leucocyte functionality, we analyzed IL-6, IFN-γ or IL-1β production by lymphocytes and monocytes after 4 hours of culture in the presence of human rIL-6 (100 ng/mL). Additionally, PMA/Ion, (50 ng/mL & 1 mg/mL) or LPS (250 ng/mL) was used as a polyclonal stimulus. Fig 2 shows that human rIL-6 did not increase the percentages of CD4+IL-6+, CD4+IFN-γ+, CD8+IL-6+, or CD8+IFN-γ+ T lymphocytes (Fig 2B, 2E, 2H and 2K). In the

**Table 2. Frequency of COVID-19 signs and symptoms.**

|  | NP-COVID-19+ (n = 15)a | P-COVID-19+ (n = 13)b | P-COVID-19- (n = 12)c | P |
|---|---|---|---|---|
| Cough | 10 | 9 | 0 | 0.001 |
| Rhinorrhea | 4 | 6 | 0 | 0.038 |
| Odynophagia | 5 | 3 | 0 | **0.092** |
| Myalgia | 10 | 8 | 0 | 0.001 |
| Arthralgia | 7 | 7 | 0 | 0.012 |
| Anosmia | 2 | 1 | 0 | **0.417** |
| Dyspnea | 11 | 6 | 0 | 0.0006 |
| Diarrhea | 3 | 1 | 0 | **0.202** |

Non-Pregnant COVID-19+ (NP-COVID-19+), Pregnant COVID-19+ (P-COVID-19+), Pregnant COVID-19- (P-COVID-19-). Fisher's exact test. Significant $p<0.05$.

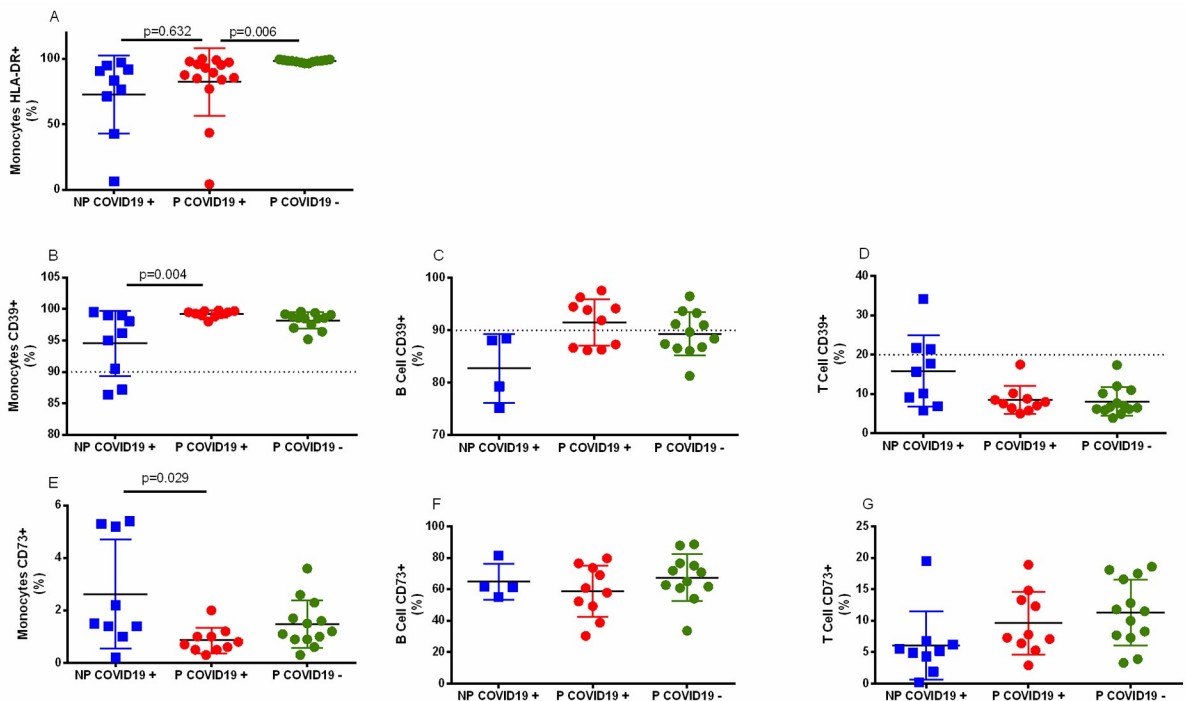

**Fig 1. Surface marker on leucocytes.** Whole blood cells were immunophenotyped as described in the methods. The results are expressed as the mean±SD. Significance value was *p*<0.05. Kruskal–Wallis and Dunn's multiple comparisons tests were calculated. Non-Pregnant COVID-19 positive (NP-COVID-19+, n = 4–9). Pregnant COVID-19 positive (P-COVID-19+, n = 10–15). Pregnant COVID-19 negative (P-COVID-19-, n = 12–13). The dotted line indicates the percentage of monocytes, B cells and T cells that constitutively express CD39 [32].

presence of human rIL-6, only 5% of the T cells expressed IL-6 or IFN-γ, a percentage similar to that of whole blood alone (Fig 2A, 2D, 2G and 2J). In addition, in response to PMA/Ion, the percentage of CD4+IL-6+ T lymphocytes was higher in the P-COVID-19+ group than in the P-COVID-19- group (Fig 2C). This response was not observed in the CD8+IL-6+ cells (Fig 2I). In addition, PMA/Ion stimulus increased the percentage of CD4+ IFN-γ+ and CD8+IFN-γ+ cells in the pregnant women with and without COVID-19 (Fig 2F and 2L), and a lower response was observed in the NP-COVID-19+ group. These differences did not reach statistical significance.

The proportion of IL-1+ monocytes or IL-6+ monocytes reached less than 10% in the groups without stimuli (Fig 2M and 2P), and human rIL-6 increased the percentage of monocytes IL-1+ and IL-6+ by 5–10%. Finally, the percentages of IL-1+ and IL-6+ monocytes after LPS challenge were similar among the groups (Fig 2Q and 2R), and after LPS stimulation, all patients reached 60–80% IL-1+ monocytes and 30–50% IL-6+ monocytes.

To determine the serum concentrations of cytokines/chemokines, collected samples were analyzed and compared among groups. The TNF-α, IL-6, MIP1b and IL-4 concentrations were higher in the P-COVID-19+ than the P-COVID-19- patients (Fig 3A, 3B, 3D and 3E). Additionally, these cytokines were at higher concentrations in the P-COVID-19+ than the NP-COVID-19+ patients; however, they did not reach a significant difference. Another chemokine, CXCL10 (IP10), showed a higher concentration in the P-COVID-19+ than the P-COVID-19- patients, although it did not reach a statistically significant difference (Fig 3C, p = 0.163). S1 Fig shows other cytokines/chemokines that showed similar concentrations

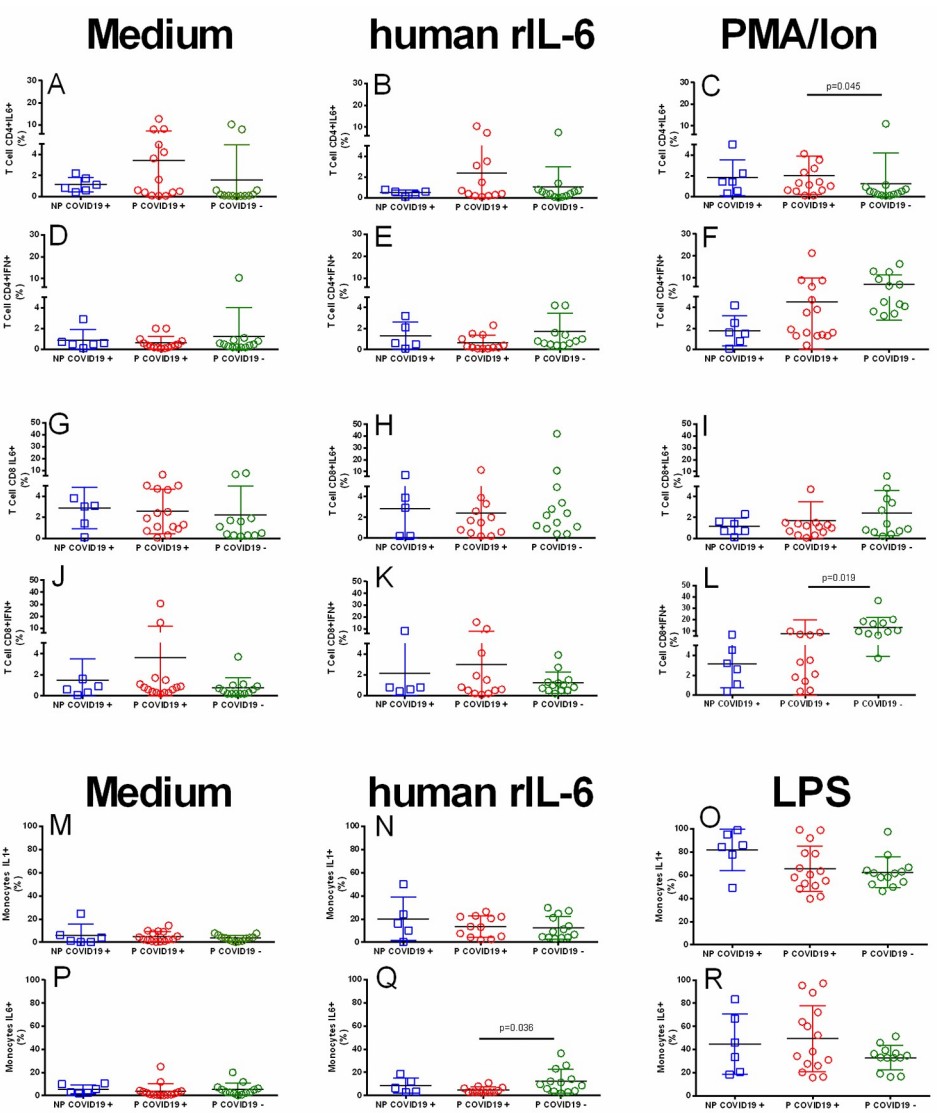

**Fig 2. Percentage of cytokine-positive leucocytes.** Whole blood cells were immunophenotyped as described in the methods. The results are expressed as the mean±SD. Significance value was $p < 0.05$. Kruskal–Wallis and Dunn's multiple comparisons tests were calculated. Non-Pregnant COVID-19 positive (NP-COVID-19+, n = 5–6). Pregnant COVID-19 positive (P-COVID-19+, n = 12–15). Pregnant COVID-19 negative (P-COVID-19-, n = 12–13).

between and among groups, including CXCL8, CCL11, CCL17, CCL2, CCL5, CCL3, CXCL9, CXCL5, CCL23, CXCL1, CXCL11, IL-17a, IFN-γ and IL-10.

To determine whether leucocytes from P-COVID19+ patients could secrete cytokines/chemokines in response to polyclonal challenge, whole blood was incubated with IL-6, PMA/ionomicin or LPS, and the supernatant was recovered after 4 hours. Fig 4 shows that in medium alone, the TNF-α, IFN-γ, CCL3, CCL4, IL-17a, CCL23, CXCL8 and IL-10 concentrations were similar among the groups (Fig 4A–4H). Human rIL-6 induced a higher concentration of TNF-α and CCL3 than medium alone (Fig 4A and 4B). Additionally, LPS induced higher concentrations of TNF-α, CCL3, CCL4, CCL23 and CXCL8 than medium alone, and these differences reached high significance (p<0.0001). Finally, PMA/ion stimulus induced higher concentrations of CCL3, CCL4, IL-17a, CCL23, CXCL-8 and IL-10 in the groups than

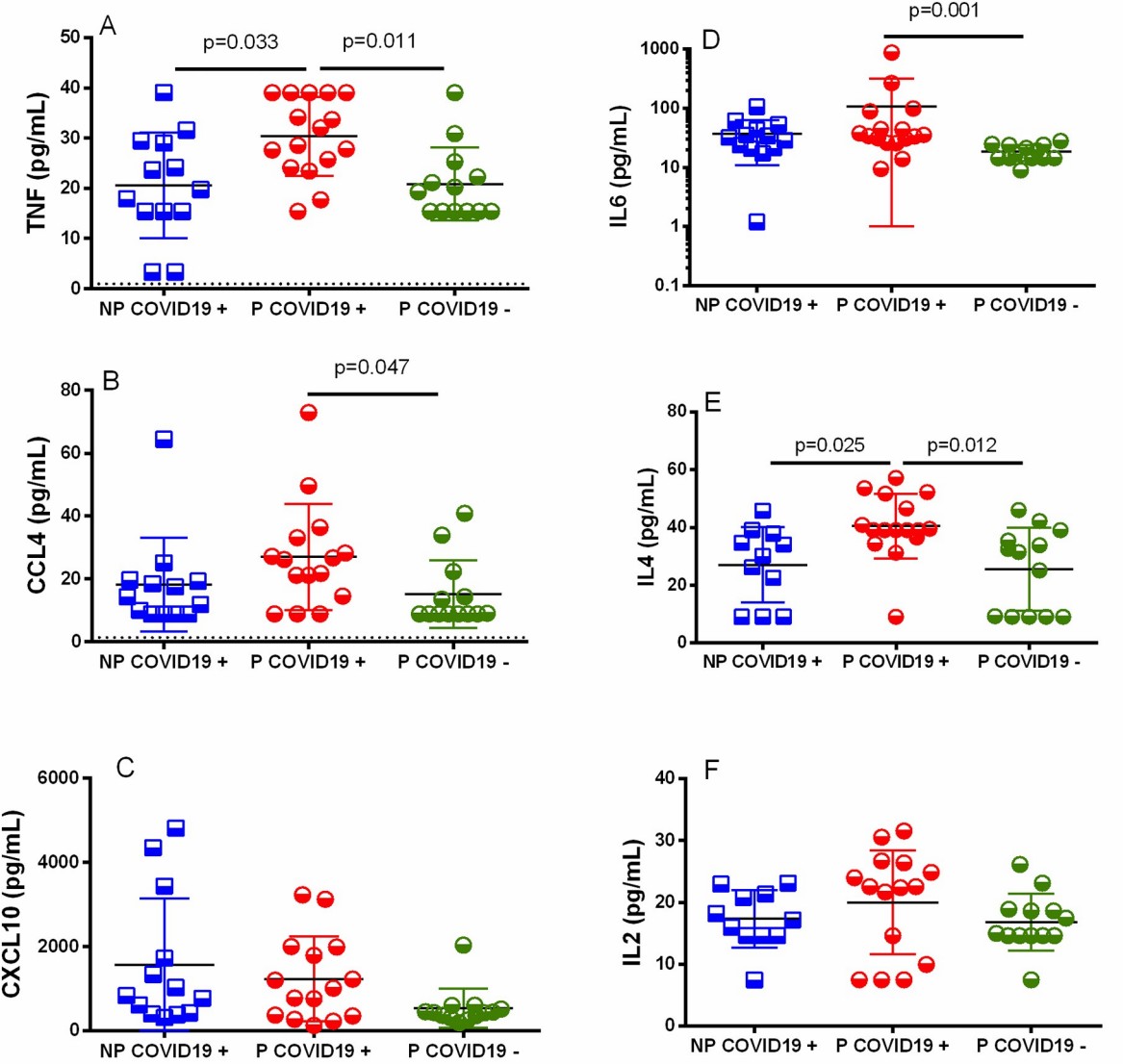

**Fig 3. Cytokine/chemokine serum concentration.** Serum was isolated, and cytokine/chemokine concentrations were determined using bead-based immunoassays as described in the methods. The results are expressed as the mean±SD. Significance value was $p<0.05$. Kruskal–Wallis and Dunn's multiple comparisons tests were calculated. Non-Pregnant COVID-19 positive (NP-COVID-19+, n = 13). Pregnant COVID-19 positive (P-COVID-19+, n = 14–15). Pregnant COVID-19 negative (P-COVID-19-, n = 13).

medium alone (p<0.0001). However, PMA/Ion did not induce an increase in IFN-γ concentration in the groups, and the lowest response was observed in the NP-COVID-19+ group. Although the IFN-γ concentration was higher in the pregnant women with and without COVID-19 than in the NP-COVID-19+ group, no statistically significant difference was achieved (both p>0.9). Additionally, S2 Fig shows that the polyclonal stimulus did not induce an increase in the concentrations of CCL11, CCL17, CCL2, CCL5, CXCL9, CXCL5, CXCL1, CXCL11, IL-4 or IL-2.

## Discussion

During pregnancy, the immune system is highly regulated. Multiple mechanisms of immune tolerance develop during gestation and favor physiological progress in reproduction [8]. Any

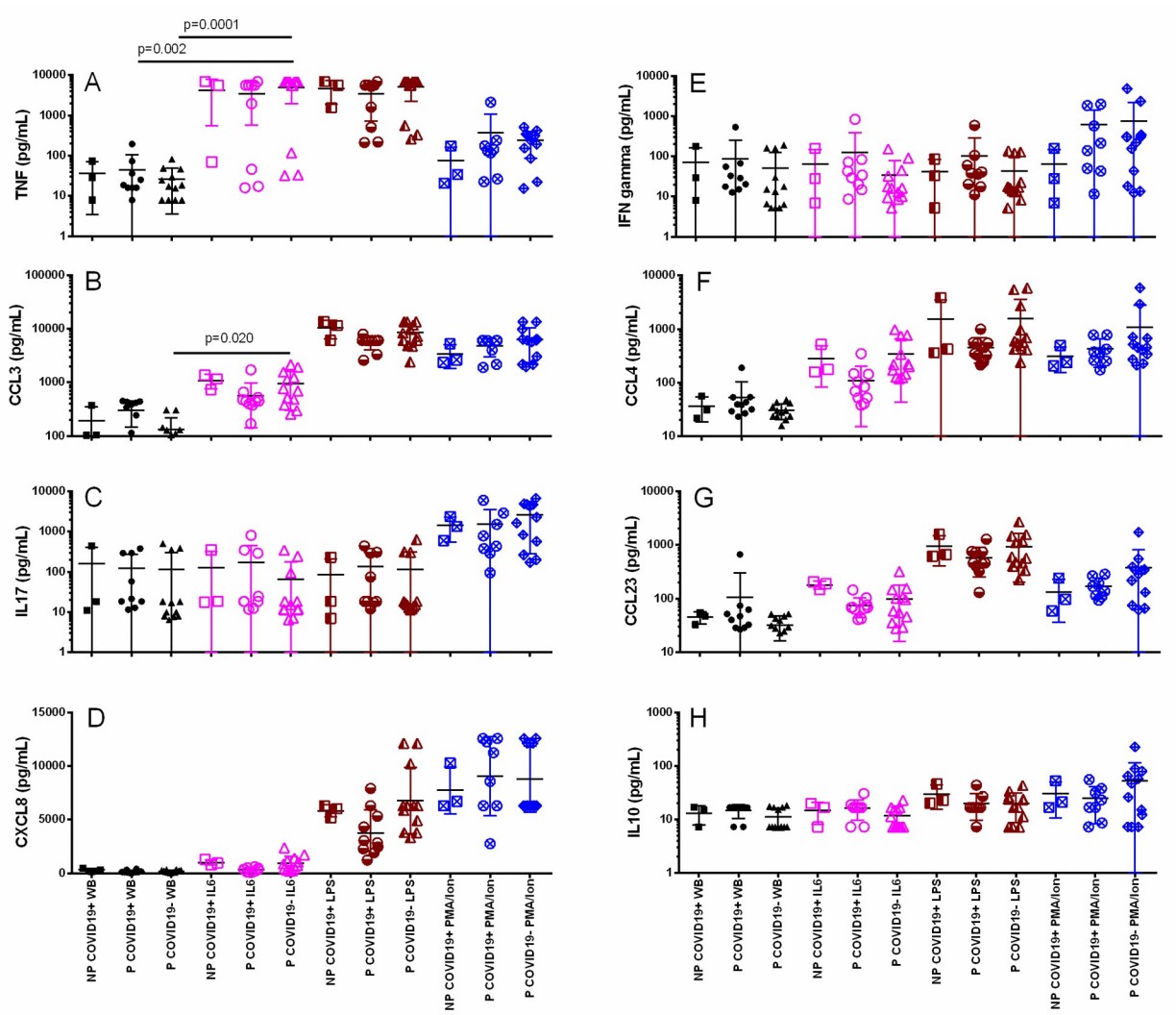

**Fig 4. Cytokine/chemokine response after 4 hours of culture with polyclonal stimulus in pregnant and nonpregnant women with or without COVID-19.** The supernatant was collected, and cytokine/chemokine concentrations were determined using bead-based immunoassays as described in the methods. The results are expressed as the mean±SD. Significance value was $p<0.05$. Kruskal–Wallis and Dunn's multiple comparisons tests were calculated. Non-Pregnant COVID-19 positive (NP-COVID-19+, n = 3). Pregnant COVID-19 positive (P-COVID-19+, n = 8–10). Pregnant COVID-19 negative (P-COVID-19-, n = 12). WB, Whole Blood.

viral infection poses a high risk of increasing fetal-maternal morbimortality partially by deregulation of cellular and humoral immune responses. Villar et al. reported a greater probability of morbidity and mortality in P-COVID-19+ than in P-COVID-19- patients [2]. However, we found no differences in clinical manifestations or laboratory values between P-COVID-19 + and P-COVID-19- patients. Additionally, we did not observe differences in several clinical features related to chronic inflammatory response (BMI/fibrinogen) or metabolic/renal status (glucose, creatinine, DHL). This perception could result from the limited number of observations. However, we observed some differences between pregnant and nonpregnant women, such as a higher serum urea concentration, a longer PT time, and a lower concentration of D-dimer in NP-COVID-19+ than in P-COVID-19+ or P-COVID-19- patients. In pregnancy, such differences occur because of the continuous and physiologic activation of the coagulation

system [33]; the longer PT time and lower D-dimer concentration in the NP-COVID-19+ con-
dition could represent the worst adaptation of the pathophysiologic response in SARS-CoV-2
infection. Accordingly, there are few reports of coagulopathy in P-COVID-19+ patients [34].
In contrast, a higher D-dimer concentration has been reported in the general population,
which correlates with a high frequency of fatal outcomes [35–37]. Interestingly, we observed
similar D-dimer concentrations in P-COVID-19+ and P-COVID-19- patients, suggesting that
in pregnant women, D-dimer concentration may not be a good predictor for severity or
thromboembolic risk [38]. Nevertheless, a larger number of observations within a longitudinal
study is needed to assess the usefulness of D-dimer concentration as a predictor in pregnant
women with COVID-19.

We observed a lower proportion of HLA-DR+ monocytes in COVID-19+ women with and
without pregnancy (Fig 1A), and this lower proportion has also been observed in septic
patients with critical conditions or fatal outcomes [39, 40]. The lower percentage of HLA-DR
+ monocytes in COVID-19 could represent the downregulation process of the immune
response by SARS-CoV-2, which helps viruses evade immunity, or could be a physiologic
mechanism that controls activation of the immune system to avoid overstimulation. More
studies are necessary to answer these questions.

An early activation marker (CD69) was also analyzed in our study. We found a similar
result for CD69+ cells in both CD4 and CD8 lymphocytes in pregnant and nonpregnant
COVID-19 patients (S2 Table). In contrast, a high percentage of CD69+ lymphocytes is
observed in pregnant patients with AH1N1 influenza, another respiratory virus challenge that
leads to unregulated inflammation in the lungs [41]. These data suggest that depending on the
virus that infects pregnant women, different leucocytes become activated, which may require
modifications in medical treatment to obtain better results.

Pregnancy is a unique immunological condition, and viral infection represents a major
challenge that could alter the immune balance. Innate and acquired responses are participants
of a highly regulated immune response in pregnancy [6, 8]. CD39 and CD73 are ectoenzymes
that sequentially metabolize ATP to adenosine, leading to an anti-inflammatory response [27,
32]. Dorneles et al. showed a higher percentage of CD4+CD39+ cells in severe COVID-19
patients than in healthy controls and a lower percentage of CD4+CD73+ cells than in controls
[30], suggesting that this could be a useful marker to follow progression in the general popula-
tion with COVID-19. Our results showed that the percentage of CD39+ or CD73+ B or T cells
was not significantly modified by the effects of pregnancy or COVID-19 infection (Fig 1).
However, we observed higher percentages of CD39+ monocytes and a lower percentage of
CD73+ cells in both pregnant women with and without COVID-19 than in nonpregnant
COVID-19+ women (Fig 1). Therefore, pregnant women may modulate inflammation
through CD39+ or CD73+ monocytes; these monocytes, as suggested for the general popula-
tion, could be a potential marker to monitor the evolution of pregnant women hospitalized for
COVID-19. Since early growth and remodeling occur in pregnancy, several opportunities to
release alarmins (such as ATP) are quite possible; in this regard, CD39- and CD73-positive
cells could be useful to maintain the regulation of inflammation. Whether this mechanism
could also be involved in limiting the inflammatory response in pregnant women with asymp-
tomatic SARS-CoV-2 infection or in mild COVID-19 has to be elucidated.

Activated leucocytes are a potential source of proinflammatory or regulatory cytokines in
the peripheral blood of COVID-19 patients. We determined the percentages of IL-1β+, IL-6
+ and IFN-γ leucocytes after 4 hours of culture with or without polyclonal stimulation. CD4
+IL-6+ or IFN-γ+, CD8+IL-6+ or IFN-γ+ lymphocytes and IL-1β+ or IL-6+ monocytes did
not reach more than 5% of circulating cells, indicating a low baseline of leucocyte producers
for these cytokines in blood. After stimulation with human rIL6, there was no significant

increase in IL-1β, IL-6 or IFN-γ lymphocyte or monocyte producers, suggesting that human rIL-6 in COVID-19 patients may not induce the synthesis of proinflammatory cytokines by circulating leucocytes. Polyclonal stimulation with LPS increases the percentage of IL-1β+ or IL-6+ monocytes, and PMA/Ion increases the proportion of CD4+IFN-γ+ and CD8+IFN-γ+ T cells, indicating that despite pregnancy or SARS-CoV-2 infection, mononuclear cells are not anergic and retain the ability to express a cytokine response. We observed a trend toward an increase in the percentage of IL-6+ monocytes and a decrease in the percentage of CD4+IFN-γ+ and CD8+IFN-γ+ lymphocytes in COVID-19 patients, indicating that mononuclear cells support a high concentration of IL-6 and a low response to IFN-γ.

Serological TNF-α, IL-6, CCL4 and IL-4 levels were higher in P-COVID-19+ than in P-COVID-19-, although TNF-α was significantly higher in P-COVID-19+ than in P-COVID-19- (Fig 3A). Interestingly, we also found the highest concentration of IL-4 in the P-COVID-19+ and reached a statistically significant difference with P-COVID-19+ (p = 0.025) and P-COVID-19- (p = 0.012). Our results indicate a similar proinflammatory profile in COVID-19+ patients with or without pregnancy and probably regulated at least partially by IL-4 in pregnant women. Additionally, syncytiotrophoblasts in the human placenta secrete vesicle-enclosed microRNAs that could limit viral infections [42]. It has been reported that Let-7 is a miRNA expressed in human cells, let-7a and let-7c inhibit the expression of IL-6, and the increase in the levels of let-7-5p and let-7-3p reduces the expression of IL-1β, IL-8, CCL2 and GM-CSF in cell lines [43]. These results open the possibility that the human placenta regulates cytokine storms in SARS-CoV-2 through miRNAs by the placenta, and more studies are necessary to probe this hypothesis. Some reports have shown that the CXCL10 concentration in serum is associated with poor prognosis in COVID-19+ patients [18]. In contrast, we observed lower CXCL10 concentrations in pregnant COVID-19+ patients than in NP-COVID-19+ patients. This suggests that the immune response in pregnancy is less severe than in the general population.

After polyclonal stimulation, the basal and leucocyte responses showed a similar production of cytokines among the groups (Fig 4), indicating that although COVID-19 peripheral leucocytes have a similar capacity to produce cytokines and suggesting that no anergy or hyperresponse is supported by SARS-CoV-2 infection in leucocytes, hypercytokinemia in COVID-19 could depend not only on leucocytes but also on an alternate source, such as endothelial cells. Human rIL-6 stimulus caused an increase in some cytokines, such as TNF-α and CCL3, indicating that IL-6 signaling favors the synthesis of some cytokines but not the entire set of cytokines related to the so-called cytokine storm. In addition, the response to IL-6 in COVID-19 seems to be similar in the presence or absence of pregnancy, indicating that pregnancy does not necessarily aggravate the proinflammatory responses (at least the leucocyte responses) in COVID-19. To explore whether the leucocyte response is limited by COVID-19, we stimulated cells with LPS or PMA/Ion, and a clear proinflammatory response with cytokines such as TNF-α, CCL3, CCL4, IL-17a, CCL23 and CXCL8 was detected in the supernatant, indicating that leucocytes are not anergic. It has been proposed that immunosuppression rather than a hypercytokine response in COVID-19 could support the pathophysiology [44]. Our results indicate that peripheral blood leucocytes from pregnant women with COVID-19 are capable of expressing a similar response to those from healthy pregnant women.

The main limitation of the present study is the limited number of patients and observations. A greater number of observations is required to reach more certain conclusions. However, the reproducibility and consistency of these results back our analysis. We propose that CD39/CD73 expression in leucocytes, mainly monocytes, could be a candidate to monitor the evolution of COVID-19. On the other hand, unlike in the general population, D-dimer concentrations in pregnant women are not necessarily a marker for severity or thromboembolic risk. Our

findings showed that the immune profile in pregnant women was similar to that in nonpregnant women when both were hospitalized for COVID-19. Nevertheless, the slight differences in peripheral leucocyte immunophenotypes suggest that pregnancy elicited pathophysiological mechanisms during COVID-19 infection that require future studies to be clarified.

## Supporting information

**S1 Table. Percentage of leucocytes after 4 hours of medium culture in pregnant and non-pregnant women with or without COVID-19.**
(TIF)

**S2 Table. Surface of activation markers on leucocytes in pregnant and nonpregnant women with or without COVID-19.**
(TIF)

**S3 Table. Cytokine-positive leucocyte percentage after 4 hours of polyclonal stimulus in pregnant and nonpregnant women with or without COVID-19.**
(TIF)

**S1 Fig. Similar cytokine/chemokine serum concentrations in pregnant and nonpregnant women with or without COVID-19.** Serum was isolated, and cytokine/chemokine concentrations were determined using bead-based immunoassays as described in the methods. The results are expressed as the mean±SD. Significance value was $p<0.05$. Kruskal–Wallis and Dunn's multiple comparisons tests were calculated. Non-Pregnant COVID-19 positive (NP-COVID-19+, n = 13). Pregnant COVID-19 positive (P-COVID-19+, n = 15–16). Pregnant COVID-19 negative (P-COVID-19-, n = 13).
(TIF)

**S2 Fig. Similar cytokine/chemokine response after 4 hours of culture with polyclonal stimulus in pregnant and nonpregnant women with or without COVID-19.** The supernatant was collected, and cytokine/chemokine concentrations were determined using bead-based immunoassays as described in the methods. The results are expressed as the mean±SD. Significance value was $p<0.05$. Kruskal–Wallis and Dunn's multiple comparisons tests were calculated. Non-Pregnant COVID-19 positive (NP-COVID-19+, n = 3). Pregnant COVID-19 positive (P-COVID-19+, n = 9–10). Pregnant COVID-19 negative (P-COVID-19-, n = 12). WB, Whole Blood.
(TIF)

**S1 Data.**
(XLSX)

## Acknowledgments

The authors extend gratitude to the staff at the Specialties Hospital, National Medical Center "XXI Century", Gynecology & Obstetrics Department in the General Hospital of Mexico "Dr. Eduardo Liceaga" and Gynecology & Obstetric Hospital No. 4 UMAE "Dr. Luis Castelazo Ayala".

## Author Contributions

**Conceptualization:** A. Cérbulo-Vázquez, E. Ferat-Osorio, B. Zavala-Barrios, R. L. Madera-Sandoval, C. López-Macías.

**Data curation:** G. L. Cabrera-Rivera, M. T. García de la Rosa, J. L. Prieto-Chávez, R. L. Madera-Sandoval, E. Salazar-Rios, M. E. Salazar-Rios, D. Serrano-Molina, A. H. Villa-nueva-Compean, R. Ramirez-Montes de Oca, F. Caldiño-Soto, L. A. Ramírez-García, O. Moreno-Álvarez, G. M. L. Guerrero-Avendaño.

**Formal analysis:** G. L. Cabrera-Rivera, R. L. Madera-Sandoval, A. H. Villanueva-Compean, R. Ramirez-Montes de Oca, F. Caldiño-Soto, L. A. Ramírez-García, O. Moreno-Álvarez, G. M. L. Guerrero-Avendaño.

**Funding acquisition:** C. López-Macías.

**Investigation:** B. Zavala-Barrios, G. L. Cabrera-Rivera, G. M. L. Guerrero-Avendaño.

**Methodology:** L. Arriaga-Pizano, G. L. Cabrera-Rivera, P. Miranda-Cruz, M. T. García de la Rosa, J. L. Prieto-Chávez, V. Rivero-Arredondo, R. L. Madera-Sandoval, A. Cruz-Cruz, E. Salazar-Rios, D. Serrano-Molina, R. C. De Lira-Barraza, A. H. Villanueva-Compean, A. Esquivel-Pineda, R. Ramirez-Montes de Oca, F. Caldiño-Soto, L. A. Ramírez-García, G. Flores-Padilla, O. Moreno-Álvarez, G. M. L. Guerrero-Avendaño.

**Resources:** C. López-Macías.

**Supervision:** L. Arriaga-Pizano, E. Ferat-Osorio.

**Validation:** L. Arriaga-Pizano, E. Ferat-Osorio.

**Writing – original draft:** A. Cérbulo-Vázquez.

**Writing – review & editing:** A. Cérbulo-Vázquez, M. García-Espinosa, J. C. Briones-Garduño, L. Arriaga-Pizano, E. Ferat-Osorio, R. L. Madera-Sandoval, M. E. Salazar-Rios, G. Flores-Padilla, C. López-Macías.

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
