## [Decision Letter · Decision Letter 0]

22 Nov 2021

PONE-D-21-29626The percentage of Monocytes CD39+ is higher in Pregnant COVID19+ than in Non-Pregnant COVID19+ patientsPLOS ONE

Dear Dr. Cerbulo,

Thank you for submitting your manuscript to PLOS ONE. After careful consideration, we feel that it has merit but does not fully meet PLOS ONE’s publication criteria as it currently stands. Therefore, we invite you to submit a revised version of the manuscript that addresses the points raised during the review process.

We look forward to receiving your revised manuscript.

Kind regards,

Eliseo A Eugenin, Ph.D.

Academic Editor

PLOS ONE

Additional Editor Comments (if provided):

Dear Dr. Cerbulo-Vasquez

Thank you for submit your manuscript to PLOSone. Please answer the comments and suggestions of the reviewers. The more important issue is the clarification of the data and correlate your data with the discussion.

Best Regards

Eliseo Eugenin

“This project was supported by the Mexican National Research Council (CONACyT), (Project No. 313494 awarded to CLM).”

We note that you have provided funding information within the Acknowledgements Section. Please note that funding information should not appear in the Acknowledgments section or other areas of your manuscript. We will only publish funding information present in the Funding Statement section of the online submission form.

“CLM awarded a Project No. 313494 by the Mexican National Research Council (CONACyT). https://conacyt.mx/

5. Please amend the manuscript submission data (via Edit Submission) to include author “C. De Lira-Barraza, A. Esquivel-Pineda, G. Flores-Padilla”

Reviewers' comments:

Reviewer's Responses to Questions

**Comments to the Author**

1. Is the manuscript technically sound, and do the data support the conclusions?

Reviewer #1: Partly

Reviewer #2: Yes

2. Has the statistical analysis been performed appropriately and rigorously? 

Reviewer #1: Yes

Reviewer #2: Yes

3. Have the authors made all data underlying the findings in their manuscript fully available?

Reviewer #1: Yes

Reviewer #2: Yes

4. Is the manuscript presented in an intelligible fashion and written in standard English?

Reviewer #1: No

Reviewer #2: Yes

5. Review Comments to the Author

Reviewer #1: Review of Manuscript for PLOS ONE

Reviewer: Riad Bayoumi

PONE-D-21-29626, entitled "The percentage of Monocytes CD39+ is higher in Pregnant COVID19+ than in Non-Pregnant COVID19+ patients".

Review:

1. The study presents the results of original research.

2. Results reported have not been published elsewhere.

3. Experiments, statistics, and other analyses are described in sufficient detail.

4. Conclusions presented are not appropriate and are not supported by the data.

a. The authors did not refer to the purinergic pathway in all sections of the MS, even though the CD39 and CD73 data were available to them.

b. In spite of available data on both CD39 and CD73 authors interpretation of their data and conclusions did not touch on immune suppression/uncontrolled inflammation by [+/-] Adenosine seen in women with a cytokine storm.

c. Several new articles on the subject have not been referred to: (Alahmadi et al, 2020: Cells);

(Alberca et al 2020 Frontiers in Immunology); (Chen et al, 2021 Signal transduction); (Dorneles et al, 2021; MedRxiv).

5. The article is presented in very poor English. It must be edited by an English native speaker.

6. The research meets all applicable standards for the ethics of experimentation and research integrity.

7. The article adheres to appropriate reporting guidelines and community standards for data availability.

Reviewer #2: The manuscript is very interesting exploring a new immunological field. In my opinion the Authors should better explain the maternal-fetal tolerance that is the key point of the current research. Please mark the suggestive clinical and laboratory characteristics.

6. PLOS authors have the option to publish the peer review history of their article (what does this mean?). If published, this will include your full peer review and any attached files.

Reviewer #1: No

Reviewer #2: **Yes: **Raffaele Falsaperla

---

## [Author Response · Author response to Decision Letter 0]

6 Jan 2022

Eliseo A Eugenin, Ph.D.

Academic Editor

PLOS ONE

Dear Doctor

Authors want to thank the journal and reviewers, your comments and suggestions really help to improve our manuscript. All points raised by the reviewers were addressed to improve the clarity of the manuscript. 

Attending PLOS ONE requirements, we did:

RE: Now the manuscript meets PLOS ONE's style requirements, including those for file naming.

2. We note that you have provided funding information within the Acknowledgements Section. Please note that funding information should not appear in the Acknowledgments section or other areas of your manuscript. We will only publish funding information present in the Funding Statement section of the online submission form. Please remove any funding-related text from the manuscript and let us know how you would like to update your Funding Statement. 

RE: We remove funding information from the manuscript. 

3. In your Data Availability statement, you have not specified where the minimal data set underlying the results described in your manuscript can be found. PLOS defines a study's minimal data set as the underlying data used to reach the conclusions drawn in the manuscript and any additional data required to replicate the reported study findings in their entirety.

RE: We upload minimal underlying data in an excel file, and verified that all patient´s information is fully anonymized.

4. Please include captions for your Supporting Information files at the end of your manuscript, and update any in-text citations to match accordingly.

RE: We include captions for supporting information files, and update any in-text citations to match accordingly.

5. Please amend the manuscript submission data (via Edit Submission) to include author “C. De Lira-Barraza, A. Esquivel-Pineda, G. Flores-Padilla”

RE: We include authors “C. De Lira-Barraza, A. Esquivel-Pineda, and G. Flores-Padilla”

Comments to the Author

6. Conclusions presented are not appropriate and are not supported by the data.

a. The authors did not refer to the purinergic pathway in all sections of the MS, even though the CD39 and CD73 data were available to them.

b. In spite of available data on both CD39 and CD73 authors interpretation of their data and conclusions did not touch on immune suppression/uncontrolled inflammation by [+/-] Adenosine seen in women with a cytokine storm.

c. Several new articles on the subject have not been referred to: (Alahmadi et al, 2020: Cells); (Alberca et al 2020 Frontiers in Immunology); (Chen et al, 2021 Signal transduction); (Dorneles et al, 2021; MedRxiv).

RE: We apologize if the first version of our manuscript expresses conclusions not appropriate and not supported by the data, we modified the text and hope that this mistake have been solved now. The changes are highlighted in yellow in the manuscript. In addition, we have cited the articles suggested by the reviewers which now appear in reference section.

7. The article is presented in very poor English. It must be edited by an English native speaker.

RE: The manuscript English editing (language, grammar, punctuation, spelling, and overall style) was performed by AJE which is one the most prestigious qualified native English-speaking editors. We include the editing certificate we received from them.

8. The manuscript is very interesting exploring a new immunological field. In my opinion the Authors should better explain the maternal-fetal tolerance that is the key point of the current research. Please mark the suggestive clinical and laboratory characteristics.

RE: We agree that the maternal-fetal tolerance is the key point of the current research, however, functional analysis was limited and we want to be prudent at this respect. We made some changes in the text that we hope could help to address this point.

Finally, while performing the manuscript revision to address the reviewers’ comments we identified an error: two patients were wrongly assigned to P-COVID-19+ group; one of them belonged to the NP-COVID-19+ group and the other to the P-COVID19- group. With the corrected groups, we performed the statistical analysis for all demographic, clinical, cytokine/chemokine, immunophenotype and functional data. Despite some p values changed in the significance value (<0.001 vs <0.05, for example), it did not change neither the significance nor the direction of the results. We clarify that the messages of the figures (even with the corrections) are the same, as well as the general conclusion. We apologize for this mistake and, in the understanding that true data is the essence of science, we really appreciate the opportunity to present you this improve and corrected version of our manuscript.

Sincerely yours,

Corresponding author:

Arturo Cérbulo Vázquez. MD. PhD

Hospital General de México “Dr. Eduardo Liceaga”

E-mail: cerbulo@unam.mx

---

## [Decision Letter · Decision Letter 1]

14 Feb 2022

The percentage of CD39+ monocytes is higher in pregnant COVID-19+ patients than in nonpregnant COVID-19+ patients

PONE-D-21-29626R1

Dear Dr.  Cérbulo-Vázquez,

We’re pleased to inform you that your manuscript has been judged scientifically suitable for publication and will be formally accepted for publication once it meets all outstanding technical requirements.

Kind regards,

Eliseo A Eugenin, Ph.D.

Academic Editor

PLOS ONE

Additional Editor Comments (optional):

Dear Dr. Cerbulo-Vazquez

Thank you for submit the response to the comments of the reviewers.

Best regards

Eliseo

Reviewers' comments:

Reviewer's Responses to Questions

**Comments to the Author**

1. If the authors have adequately addressed your comments raised in a previous round of review and you feel that this manuscript is now acceptable for publication, you may indicate that here to bypass the “Comments to the Author” section, enter your conflict of interest statement in the “Confidential to Editor” section, and submit your "Accept" recommendation.

Reviewer #1: All comments have been addressed

2. Is the manuscript technically sound, and do the data support the conclusions?

Reviewer #1: Yes

3. Has the statistical analysis been performed appropriately and rigorously? 

Reviewer #1: Yes

4. Have the authors made all data underlying the findings in their manuscript fully available?

Reviewer #1: Yes

5. Is the manuscript presented in an intelligible fashion and written in standard English?

Reviewer #1: Yes

6. Review Comments to the Author

Reviewer #1: (No Response)

7. PLOS authors have the option to publish the peer review history of their article (what does this mean?). If published, this will include your full peer review and any attached files.

Reviewer #1: No

---

## [Editor Report · Acceptance letter]

1 Mar 2022

PONE-D-21-29626R1 

The percentage of CD39+ monocytes is higher in pregnant COVID-19+ patients than in nonpregnant COVID-19+ patients 

Dear Dr. Cérbulo-Vázquez:

I'm pleased to inform you that your manuscript has been deemed suitable for publication in PLOS ONE. Congratulations! Your manuscript is now with our production department. 

Kind regards, 

on behalf of

Dr. Eliseo A Eugenin 

Academic Editor

PLOS ONE